# Health and Environmental Risks of Residents Living Close to a Landfill: A Case Study of Thohoyandou Landfill, Limpopo Province, South Africa

**DOI:** 10.3390/ijerph16122125

**Published:** 2019-06-15

**Authors:** Prince O. Njoku, Joshua N. Edokpayi, John O. Odiyo

**Affiliations:** 1Department of Ecology and Resource Management, University of Venda, Thohoyandou 0950, South Africa; 2Department of Hydrology and Water Resources, University of Venda, Thohoyandou 0950, South Africa; Joshua.edokpayi@univen.ac.za (J.N.E.); John.odiyo@univen.ac.za (J.O.O.)

**Keywords:** environmental risk, landfill site, perception, public health, waste disposal

## Abstract

The by-products of solid waste deposited in a landfill has adverse effects on the surrounding environment and humans living closer to landfill sites. This study sought to test the hypothesis that the deposition of waste on landfill has an impact on the surrounding environment and residents living closer to it. This was achieved by evaluating the perception of the respondents drawn from people living close (100–500 m) and far (1–2 km) from the landfill site, concerning environmental issues, health problems, and life satisfaction. Results from the study showed that 78% of participants living closer to the landfill site indicated serious contamination of air quality evident from bad odours linked to the landfill site. Illnesses such as flu, eye irritation and weakness of the body were frequently reported by participants living closer to the landfill than those living far from the landfill. More than half of the participants (56%) living closer to the landfill indicated fear of their health in the future. Thus, the participants living closer to the landfill site were less satisfied with the location of their community with respect to the landfill, than those living far from the landfill site. Therefore, the need for a landfill gas (LFG) utilisation system, proper daily covering of waste and odour diluting agents are necessary to reduce the problems of the residents living closer to the landfill site.

## 1. Introduction

Landfills are a major contributor to the world’s anthropogenic greenhouse gas (GHG) emissions because an enormous amount of CH_4_ and CO_2_ are generated from the degradation process of deposited waste in landfills [1]. Landfill operation is usually associated with contamination of surface and groundwater by leachate from the landfill (mostly if the landfill lacks adequate liners), pungent odour, loud disturbing noise from landfill bulldozers, bioaerosol emissions; volatile organic compounds [2,3,4,5,6,7]. The storage of leachate in open lagoons can influence the levels of odours experienced in a landfill site. Residents living close to landfill sites have shown concern due to several hazardous pollutants emanating from landfill operations [8]. Some other pollutants associated with deposition of waste on landfills include litter, dust, excess rodents, unexpected landfill fires, etc. [9,10,11]. The factors that influences the by-product or emissions from landfills include the kind and quantity of waste deposited, the age of the landfill, and the climatic conditions of the landfill sites. Complex chemical and microbiological reactions within the landfill often lead to the formation of several gaseous pollutants, persistent organic pollutants (such as dioxins, polycyclic aromatic hydrocarbons), heavy metals and particulate matter [8,10,11,12].

The continuous inhalation of CH_4_ by humans can cause loss of coordination, nausea, vomiting and high concentration can cause death [13,14,15]. Acidic gases like nitrogen dioxide, sulphur dioxide, and halides have harmful effects on the health and environment when introduced [16]. Studies have shown that when nitrogen dioxide and sulphur dioxide are inhaled or ingested by humans, symptoms such as nose and throat irritations, bronchoconstriction, dysproca and respiratory infections are prevalent, especially in asthmatic patients. These effects can trigger asthma attacks in asthmatic patients [13,17,18,19]. In addition, high contact of NO_2_ by humans increases the susceptibility to respiratory infections [17]. Furthermore, when these acidic gases reach the atmosphere, they tend to acidify the moisture in the atmosphere and fall down as acid rain. Phadi et al. [20] identified that sulphur dioxide has harmful effects on plant growth and productivity. In addition, humans are at the risk of reduced lung function, asthma, ataxia, paralysis, vomiting emphyserra and lung cancer when heavy metals are inhaled or ingested. Illnesses like, high blood pressure and anaemia have been shown to be caused by heavy metal pollution [17,21,22]. Additionally, when in contact in high proportions, heavy metals affect the nervous system which causes neurotoxicity leading to neuropathies with symptoms like memory disturbances, sleep disorders, anger, fatigue, head tremors, blurred vision and slurred speech. It can also cause kidney damage like initial tubular dysfunction, risk of stone formation or nephrocalcinosis, and renal cancer. When humans are exposed to a high amount of lead, it can cause injury to the dopamine system, glutamate system and N-methyl-D-Asphate (NMDA) [17,21,22,23].

Landfills generate different kinds of trace toxic elements which include carbon monoxide, hydrogen sulphide, xylene, dioxin, etc. Toxic organic micro pollutants also include polychlorinated dibenzo-para-dioxins and polychlorinated dibenzofurans (PCDDs and PCDFs) which are all called dioxins and polycyclic aromatic hydrocarbons (PAHs). Dioxin can be formed from the presence of chlorine-containing substances in the landfill and from landfill fire which is harmful to human health [10,17,24,25]. Dioxin has been linked with increase in mortality rate caused by ischemic heart disease, when ingested by humans [23]. PAHs are considered to have potential carcinogenic properties when in contact with humans which could lead to a tumour of the lungs, skin cancer and deficiencies on other parts of the body [4,17,24]. When humans inhale particulate matter, studies have shown that it leads to lining inflammation, systemic inflammatory changes and blood coagulation which can further lead to obstruction of blood vessels, angina and myocardial infraction [17]. A study conducted in a Turkish landfill, on the health risk assessment of BTEX (Benzene, Toluene, Ethylbenzene, and Xylene) emissions on landfill workers in the area shows that BTEX did not pose a health threat to the landfill workers, because the mean concentration of BTEX measured in the landfill was not sufficient and was lower than the United States Environmental Protection Agency (USEPA’s) generally acceptable excess upper-bound lifetime cancer risk of one in 10,000. However, the author noted that landfill effects on humans directly depended on the type of pollutants and the duration of exposure to the people [24].

Hydrogen sulphide (H_2_S) is a colourless and highly flammable gas. It has an odour of rotten egg and contributes immensely to the odour emissions experienced from landfill sites. It is formed when high sulphate containing compounds (like gypsum and plasterboard) are mixed with the degradable waste in the landfill site. When humans are exposed to high levels of H_2_S it could lead to malfunction of the central nervous system and respiratory paralysis [26].

Waste management has been closely associated with biological hazards. The decomposition of waste materials in the landfill; vehicle exhaust fumes and favourable weather condition can lead to the formation of bioaerosols and biological agents such as fungi, bacteria and volatile compounds (like endotoxins, β(1-3)-glucans and mycotoxins) [27,28]. Exposure to bioaerosols has been implicated with various respiratory health diseases which can provoke inflammation of the airways. Several studies have shown that occupational risk of waste handlers and landfill workers are high when compared to others [27,29,30]. Cancer and other respiratory allergies have been reported by communities living closer to landfill sites. Endotoxins are the most powerful proinflammatory component present in bioaerosols, which are components on the cell wall of Gram-negative bacteria. Heldal et al. [27], showed that the exposure to low concentration of endotoxins to waste collectors and compost workers can cause an inflammatory response to the upper airways through neutrophil activation and the release of cytokines such as IL6 and IL8 and TNF-alpha. In addition, Gladding et al. [29] showed that workers exposed to higher amounts of endotoxin and (1→3) -β-D-glucan had an increased risk for respiratory diseases as compared to others with lesser exposure. Most studies focused on biological risk association with waste and landfill workers because of their close proximity to the biological agents over time, therefore, this can be an indication of the possible health risk of people living closer to landfills.

Previous research shows that people living closer to landfill sites suffer from medical conditions such as asthma, cuts, diarrhoea, stomach pain, reoccurring flu, cholera, malaria, cough, skin irritation, cholera, diarrhoea and tuberculosis more than the people living far away from landfill sites [31,32,33,34,35,36]. The causes of the health problems are as a result of continuous exposure to chemicals; inhalation of toxic fumes and dust from the landfill sites. Additionally, a review on the “residential proximity to environmental hazards and adverse health outcomes” showed a significant correlation between residential proximity to environmental hazards and adverse health outcomes especially risks for central nervous system defects, congenital heart defects, oral defects, low birth weight, cancer, leukaemia, asthma, chronic respiratory symptoms, etc. The author noted that although residents living closer to the landfill appear to be more prone to adverse effects of health outcomes, the proximity does not equate to the individuals’ level of exposure [36]. The health hazard is dependent on the level of exposure of the residents to the pollutants and concentration of the pollutants. Landfill proximity to residents will also have significant effects on property value in the area [37,38,39,40].

Despite the proliferation of the harmful effects in recent years, not much research on health and environmental impacts on the residents living closer to landfill sites has been conducted in many landfills situated in rural and peri-urban centres in South Africa. Though, Bridges et al. [34] conducted a study in comparison of adverse effects of incinerators and landfill emissions on health. The study did not consider the environmental and economic risk and impact associated with landfill pollutants. Therefore, this study posed several relevant questions which are yet to be addressed. These questions include; (a) are there major social-economic differences between the residents living closer to the landfill and residents living far from the landfill? (b) Do the residents living closer to the landfill find the landfill’s characteristics very disturbing compared to those living far from it? (c) Do the residents living closer to the landfill suffer from some specific illnesses more than the residents living far from the landfill? (d) What is the perception in view of the community life satisfaction between residents living closer to the landfill site and residents living far away from it? This study was therefore aimed at investigating and providing answers to the above research questions.

## 2. Materials and Methods

Thohoyandou landfill is situated very close to the residential areas at approximately 100 m away. Therefore, this study sought to find out the perceptions of health impact and the way of life for residents living closer to Thohoyandou landfill. Firstly, a reconnaissance survey was conducted around the landfill site to identify the number of households and other functional institutions located in the area. It was observed that the community was located approximately 100 m away from the landfill. Therefore, the study focused on residents living approximately 100 m to 2 km away from the landfill. The households living closer to the landfill were identified to be approximately 100 households, with an average of four people per household [41]. Then, the participants of the study were strategically identified based on how long they have lived in the community. This led to 100 people identified as the sample size.

According to Brewer [42], stratified random sampling technique was adopted to identify approximately 50 participants for the study, who lived approximately 100 to 500 m, and 50 participants were identified as the control for the study who lived within 1 to 2 km away from the landfill.

A landfill operator manager and three university students from the University of Venda, South Africa were trained and recruited for data collection. A five-page questionnaire was pretested with 10 participants to identify errors and limitations of the survey tool. Furthermore, after adjustments of the questionnaire, the questionnaires were administered to a total of 100 participants (50 participants—people living closer to landfill; while 50 participants—people living far away from the landfill). At the start of the administration of questionnaires, the majority of the residents were willing to corporate and participate in the study. Additional information, suggestions and recommendations were also given to the researchers by the participants based on the environmental challenges faced by the community. During the fieldwork, four households expressed scepticism and refused to participate in the study. Two other households expressed less concern because they felt the study was not beneficial to them as they were not house owners and not fully responsible for the environmental issues in the community.

Topical issues on the perception of neighbourhood problems, the significance of environmental problems, most frequently experienced sickness and life in general in the community were identified. The questions asked concerning environmental problems were coded as (1) serious, (2) a fairly serious problem and (3) not a serious problem. The participants were provided with seven environmental issues facing the community which included disposal of solid waste, garbage, and litter in the street, unwelcome location of the landfill, air pollution, bad odour, water pollution, noise pollution and dust. An independent sample *t*-test analysis was conducted to identify the difference in mean, and the significance of results obtained from both communities (Appendix A).

In addition, the participants were presented with possible illnesses associated with complaints of people living closer to the landfill site. The participants were asked to indicate whether they or any member of their family experienced each of the identified illnesses frequently (1), fairly frequently (2) or not frequently (3). The data acquired from the field study were analysed with the aid of Statistical Package for the Social Sciences version 25 developed by International Business Machines Corporation, Armonk City, NY, USA. Figure 1 shows the location of the study area.

## 3. Results and Discussion

The social and demographic characteristics of the respondents were identified to understand the social and economic characteristics between the two communities. Table 1 shows the results obtained from the study.

Table 1 shows that there were more female than male participants in both communities due to the availability and the readiness of the female respondents to participate in this study. Participants aged from 21 to 30 years were the most dominant, though participants aged 31–40 years were equally most dominant for participants living closer to the landfill. The participants were mostly part-time workers. However, most of the participants were in high school and tertiary institutions and lived more than 5 years in the community.

### 3.1. Perception of the Significance of Environmental Problems Faced by the Community

#### 3.1.1. Disposal of Solid Waste

This indicates the rate of disposition of solid waste in the landfill and considering how serious disposal of Municipal Solid Waste (MSW) activities influence the state and wellness of the people. Table 2 indicates the comparison of participants living closer to the landfill site (CL) and participants living far away from the landfill site (AL) with regards to the significance of the impact of different landfill characteristics on both communities. A total of 70% of participants living closer to landfill site indicated that deposition of MSW in Thohoyandou landfill is a serious problem, whereas 12% of respondents living far from the landfill indicated the same problem. Furthermore, 10% of the participants living closer to the landfill site indicated that the deposition of MSW in the Thohoyandou landfill is not a serious problem to them, whereas 62% of participants living far from the landfill said the same. These responses were as a result of the physical and unpleasant presence of the landfill in the CL community. Cross tabulation between the years of participants who lived in the CL community and the significance of deposition of solid waste shows that for all the different years, solid waste disposal was a serious problem (Figure 2). This study agrees with other studies which have shown that the significant impact of deposition of MSW in landfills located in close proximity to residential areas causes negative effects to the people and the environment [24,32,33,35,36,40].

#### 3.1.2. Garbage and Litter on the Street

A total of 30% of the participants living closer to the landfill indicated that garbage and litter in the surrounding CL community was a serious problem (Table 2), while 26% of participants living far from the landfill indicated the same problem. Meanwhile, 50% of the participants living closer to the landfill indicated that the flow of garbage and litter into the community was not a serious problem to them while 12% of participants living far from the landfill indicated the same. Fifty-six percent (56%) of participants living far from the Thohoyandou landfill indicated that garbage and litter on the street were a fairly serious problem. From the responses, for both communities, garbage and litter seemed not to be a serious problem and this could be attributed to the fencing and constant covering of the MSW deposited in the landfill. A cross tabulation between the duration of years, the participants lived in the CL community and the seriousness of the problem (garbage and litter) indicates that participants who lived less than 1 year, 6–10 years and 11–20 years had high percentages indicating the impact of garbage and litter on the CL community as not a serious problem (Figure 3). Participants that lived within 5 years in the area indicated that it was a serious problem, which is possibly because these participants could have spotted several litters and attributed it to the landfill. This might not be the case, because the majority of the participants indicated that litter on the streets was not a serious problem. In addition, it is possible the litter could have come from dustbins of residents, passers-by and poor waste management system in the community. Adeola [33] indicated that garbage and litter on the streets were major problems encountered by the participants living closer to the landfill when compared to results derived from the participants living far from the landfill site. Sankoh et al. [35] conducted a study on the environmental and health impacts of the solid waste dumpsite in Freetown Sierra Leone. The study showed that the presence of the dumpsite increased the amount of filth, garbage and litter in the nearby community. Additionally, Fitaw and Zenebre [43], conducted a study in Addis Ababa city on the assessment of landfills in the city, the study showed that blowing litter from landfills have been found to be prevalent in areas closer to landfills and are easily carried to nearby residents by wind and has negative effects on the health of residents. Therefore, this shows that a controlled system of solid waste deposition and other precautionary measures are very important to achieve a cleaner environment for communities residing next to a landfill [33,36].

#### 3.1.3. Unwelcome Location of the Landfill

This indicates the suitability and acceptance of the landfill site by the participants from the community. Table 2 shows that 78% of the respondents living closer to the landfill site indicated the unsuitability of the presence of the landfill to them, whereas, 90% of participants living far from the landfill site felt that the site was fine. The cross tabulation in Figure 4 shows that all participants that have lived from less than 1 year to 20 years indicated that the landfill should not be situated closer to their homes, possibly because of the long-term risk associated with it. Studies have shown that residents living closer to the landfill site do not like the idea of the landfill’s location in close proximity to their homes because of its negative impact on their communities [3,32,34,36]. Bridges et al. [34] and Sankoh et al. [35] further showed that the exposure of participants living at least 2 km away from a landfill causes health and environmental effects when compared to participants living far from a landfill site. Thus, the findings above on the suitability of the location of landfill agree with the results found in the studies mentioned.

#### 3.1.4. Air Pollution and Bad Odour

Air pollution and bad odour have been found by many scholars to be synonymous to landfill operations. This shows the seriousness of air pollution and bad odour emanating from the landfill into the community. Table 2 shows that 78% of participants living closer to the landfill site indicated serious contamination of the air quality and the fact that they often experience a bad odour which they believe is from the landfill site. However, 16% of the participants living far from the landfill indicated serious contamination on air quality and bad odour, thus the majority of the participants living far from the landfill indicated a better air quality devoid of smell or odour. The cross tabulation between the years of participants living in the CL community and air pollution with bad odour in the community was recorded in Figure 5. The figure shows that all participants that lived for less than 1 to 20 years in the CL community indicated serious contamination in air quality with bad odour from the landfill. It was more pronounced or taken more seriously by participants who have lived longer, up to 20 years, in the CL community. Bouvier et al. [38] showed that residents living closer to landfill experienced higher contamination of air quality than residents living far from the landfill site. Vrijheid [32] identified that some components of landfill gas (LFG) like hydrogen sulphide are key contributors to odour emanating from a landfill site. Air pollution and bad odour are as a result of poor management of the landfill by landfill operators like proper compression of waste deposited in the landfill and lack of collection and utilisation of LFG emissions. However, the pungent odour and air pollution can be minimised by a proper daily covering of solid waste immediately when it is deposited in the landfill; the use of a diluting agent which suppresses bad odour from the landfill; and the collection and utilisation of the LFG emitted from the landfill. The electronic nose technique analysis by Xiangzhong [44] showed that the odour emanating from the landfill and its boundaries were similar to the odour experienced from waste sludge, residential waste and construction waste.

Sakawi et al. [4] showed in their study that about 83.7% of their respondents indicated that bad smell from landfill has affected the tranquillity and quality of life. In addition, 80.5% of participants indicated that bad odour was associated with their current bad health. The study indicates that the peak of malodour is experienced at night forcing residents to close windows and doors, thus not enjoying cross ventilation at home. Rainfall, wind direction, and intensity increased the intensity of odour emanating from the landfill. De Feo et al. [5] ascertained how participants living closer to the landfill perceive odour and local pollution. The study showed that fewer residents living closer to the waste facility complained that the facility contributed to local degradation and odour. However, the study showed that monetary compensation was given to the residents; this further influenced their perception towards odour effects from the landfill. Additionally, in 2003 during the operating year of the waste facility, the residents complained heavily of rotten egg odour coming from the landfill and it was increasing as the years went by. However, in 2009, after the closure of the waste facility residents did not complain of odour.

#### 3.1.5. Water and Noise Pollution

In the context of this study, water pollution indicates the presence of polluted water in the community. In addition, noise pollution indicates the level of noise in the community. Table 2 shows that 64% of participants living closer to the landfill indicated that the water supply was clean, while 42% of participants living far from the landfill site indicated that the water was clean. Therefore, the tap water supplied to both communities could be from a different source and not from the groundwater close to the landfill. Cross-tabulation of water and noise pollution with the duration of residents living closer to the landfill site was used to analyse the perceptions of participants who lived in different years in the CL community and how their perceptions influence the results. The cross-tabulation between the years the participants lived and water pollution shows that most participants for all age groups indicated that water pollution is not a regular problem encountered by them (Figure 6). Studies have shown that it is inevitable for landfills not to contaminate groundwater, as leachate percolates into groundwater through cracks of membranes (for sanitary landfills) and contaminates it, because of high bacteria content [6,7,11,32,45]. This study did not carry out laboratory analysis of groundwater in the area, however, the source of drinking water in the vicinity was provided by the Municipality.

Furthermore, 58% of participants living closer to the landfill indicated that there is no form of noise pollution, while 22% of participants living far from the landfill indicated the same. However, most of the participants living far from the landfill (52%) indicated noise pollution as a fairly serious problem for them. Noise can be generated from different sources and not necessarily from the landfill. Although, it is quite impossible not to notice the heavy trucks and bulldozers in a landfill, the Thohoyandou landfill lacks the adequate number of bulldozers and heavy trucks in the landfill due to lack of funds. On the day of our visit to the landfill, there were complaints about the bulldozers for daily covering of waste not functioning for several months. However, after some months when we went back to the landfill, some bulldozers were functioning. In addition, incoming waste trucks contribute some noise pollution in the landfill but not enough to cause significant pollution to the nearby residents.

Figure 7 indicates that most participants for all the age groups indicated that noise pollution is not a serious problem for them. This study is consistent with other studies that have been conducted [11,37]. Reichert et al. [37] showed that blowing trash and truck noise was the least significant problem when compared to other environmental factors. A study showed that during landfill operations that residents were a little concerned about noise pollution [11].

#### 3.1.6. Dust

Table 2 shows that 40% of the participants living closer to the landfill indicated that dust was a serious problem to them, while 4% of participants living far from the landfill site indicated that the emission of dust particles in the atmosphere was a serious problem. However, most of the participants living far from the landfill site (60%) indicated that the emission of dust particles to the atmosphere was not a serious problem for them. This shows the significance of dust particles in the atmosphere.

Figure 8 shows cross tabulation between the duration the participants lived in the CL community and dust particles in the atmosphere. All participants that lived less than one year and up to 5 years in the community indicated the emission of dust particles in the atmosphere is not a serious problem to them. However, participants that lived long in the community from 6 years up to 20 years indicated dust percolation as a serious problem in the community. Thus, it takes time for participants to experience serious dust emissions in the atmosphere. Studies have shown that dust particles from landfills have been a major concern in communities [10,11,34]. Dust emissions from landfills can be controlled by the continuous spraying of water on the soil; fan-driven misting system; mixing of X-Hension pro with water and spray on the soil; dust destroyers; etc. [10,46,47,48]. The use of spraying of water on the ground and any other technique has not been adopted in the Thohoyandou landfill and is therefore recommended.

Table 2 thus summarizes that the participants in the CL community experienced serious environmental problems with respect to the disposal of solid waste, unwelcome location of the landfill and air pollution with odour. Garbage and litter, and water and noise pollution were perceived not to be serious problems by the participants in the CL community. The participants in the AL community, however, showed lesser serious environmental problems compared to the participants in the CL community. All the environmental problems highlighted were perceived not to be serious problems except noise pollution, garbage, and litter on the street which posed some problems to the participants in the AL community.

The results show that more undesirable environmental conditions posed very serious problems for the participants in the CL community than the AL community. Specifically, disposal of solid waste; unwelcome location of landfill; and air pollution with bad odour; were considered major threats.

A *t*-test was employed to access whether the differences noted between the ratings on the significance of the environmental problem by the two locations were statistically significant (*p* < 0.05). Table 2 shows that the differences were found to be statistically significant for all seven variables in both communities. Figure 9 shows the summary in the graphical representation of the respondent’s rating living in both communities in terms of the seriousness of each of the environmental characteristic.

### 3.2. Perception of Most Reported Illnesses Encountered by the Participants in Both Communities

Respondents living close to the landfill reported that breathing disorders are frequently (24%) and fairly frequently (34%) problems they experience in the CL community. Similarly, respondents from the AL community reported that breathing disorders are frequently (10%) and fairly frequently (24%) experienced. Various studies have also shown that residents living closer to a landfill site are more prone to respiratory diseases as supported by this study [8,32,33,34,35]. Respiratory diseases and breathing disorders can be caused by bioaerosols and biological agents released from landfill sites [27]. Apart from biological agents and volatile organic compounds released from landfill sites, emissions from cars, trucks and bulldozers used in the landfill can also contribute to emissions from the landfill site [28]. Such emissions have been reported to be harmful to human health [27,28,29,30]. It is also not surprising to note that respondents living far from the landfill site also recorded respiratory diseases which were commonly experienced. Air pollution as a result of emissions from cars, biomass burning and bricks making are common anthropogenic activities in the study area and could be responsible for reported cases in the AL community. Brick making and biomass burning releases particulate matter (PM_10_ and PM_2.5_) and various volatile organic compounds that have been implicated in respiratory diseases [49,50,51,52].

A study on exposed traffic policemen to outdoor air pollution showed that the percentage of participants with a diagnosis of allergy was higher in the exposed traffic policemen than in the control [52]. Additionally, Heinrich and Wichmann [49] concluded that traffic related air pollutants can lead to mortality risk, particularly in relation to cardiopulmonary causes. The result also agrees with previous studies which shows that breathing disorders, shortness of breath and respiratory diseases are major health problems associated with landfill emissions and have continued to increase over the years [53,54,55,56].

Table 3 indicates that residents living in both communities reported the same frequent level (18%) of cancer illness. Fairly frequent cancer levels were reported at 10% and 12% for CL and AL communities, respectively.

Furthermore, illnesses like flu, eye irritation and weakness of the body were frequently reported by participants living closer to the landfill than participants living far from the landfill (Table 3). Most participants living far from the landfill indicated that they did not experience these illnesses very often. Therefore, we can conclude that there is a higher risk of most of these illnesses to be attributed to the landfill, but it is also imperative to know that these illnesses could also be contracted from various other sources. Though headache was more frequent in the CL community (38%) than AL community (20%), the latter community showed a higher percentage (54%) for fairly frequent which is an indication of significant impact.

Some illnesses recorded in this study like back pain, skin disorders, hearing impairments and asthma were indicated by most participants living closer to the landfill as not often experienced. Likewise, most participants living far from the landfill did not experience these illnesses often, except for asthma. Studies have established that cancer is an illness experienced by people living closer to a landfill or waste dump [32,33]. Similarly, the Health Protection Agency [13], showed that in several epidemiological studies performed by different scholars showing the relationship of cancer and landfill sites, cancer was a relatively complex illness to identify because of inadequate evidence to back up the claim of increased risk of cancer to communities living closer to landfill sites. Similarly, the review of Jarup et al. [57], by Small Area Health Statistics Unit (SAHSU) in 2011, showed that there was no excess risk of cancer in a people living closer to the landfill site [13].

Table 3 thus showed that the participants living closer to the landfill site reported some illnesses more often than participants living far from the landfill site. Figure 10 shows the graphical representation of the comparison of the frequency of reporting the selected illnesses in both communities. A *t*-test analysis was conducted for the most reported illnesses to understand the significance of the difference of results obtained and the result’s significance (Appendix B). Seven out of 11 health problems were statistically significant, that is breathing disorders, flu, eye irritation, weakness of the body, back pain, coughing and tuberculosis, and asthma.

### 3.3. Perception of Most Disturbing Landfill Site Characteristics

This study highlights eight disturbing characteristics, which are commonly associated with landfill sites. The participants were asked to rate the landfill characteristics based on a scale of (1) disturbing, (2) fairly disturbing or (3) not disturbing to the participants living in both communities as shown in Table 4.

Fear of future health indicates the anticipated health issues that will arise in the future based on the current effects. Table 4 shows that 56% of the participants living far from the landfill site feel that their health will be fine in the future. However, 56% of participants living closer to the landfill indicated that fear of their health in the future was a disturbing issue, while 24% of participants living far from the landfill indicated the same. This result could be attributed to the physical presence of the landfill, odour and possible fear of accumulated intake of gaseous emissions from the landfill. Similarly, Adeola [33] made a comparison of participants living closer to a landfill and far from a landfill concerning how they feared their health in the future. The study concluded that more participants living closer to the landfill site feared for their health in the future than participants living far from the landfill site.

The viability of properties in the area was also assessed. Results in Table 4 show that 54% of the participants living closer to the landfill site indicated the difficulties in the sale of the property, but 66% of the participants living far from the landfill indicated the property sale as a good business in the community. The respondents targeted to give their views concerning property sales were mainly house owners, tenants and elderly participants that had lived in both communities for a long time. Adeola [33], in a study, experienced that participants living closer to the landfill site could not sell the property as much as participants living far from the landfill site. Property buyers could be sceptical on the purchase because of close proximity of the property to the landfill site.

Additionally, other external factors on landfill characteristics like friend’s unwillingness to visit, desirable business enterprise staying away and landfill stigmatisation, show that most participants living closer to the landfill site felt that these external factors were disturbing to them. Participants living far from the landfill felt their external factors were not disturbing to them except for landfill stigmatisation. However, these communities are still developing and might still lack some desirable businesses and poor rent for properties. Rodents and mosquitoes were indicated to be more prevalent with participants living closer to the landfill than participants living far from the landfill. Therefore, some participants close the doors and windows of their houses regularly to avoid mosquitoes and rodents. Figure 11 shows the participants’ ratings on how disturbing the external factors of the landfill characteristics were to them. Thus, the participants living closer to the landfill site rated all the external landfill characteristics mostly as disturbing to them. However, the participants living far from the landfill site ranked most of the external landfill characteristics as not disturbing to them. The overall results show that the CL community was more disturbed by the external landfill characteristics than the AL community. Studies have shown that the presence of landfill in close proximity to properties reduces the values of these properties [38,39,40,58]. Seok Lim and Missios [59] indicated that the introduction of larger landfills has more impacts on property value than smaller landfills. However, according to Bouvier et al. [38], some property value depend on the buyers in question. If the buyers are not concerned about the effect of the landfill and only interested in the property, then they may pay a substantial amount for the property.

A *t*-test was used to test whether the differences noted between the ratings of the disturbances of the external landfill characteristics were statistically significant (*p* < 0.05) (Appendix C). The CL community rated the landfill characteristics more disturbing than the AL community. Appendix C shows the differences were found to be statistically significant for all the external landfill characteristics.

### 3.4. Perception of Life Satisfaction Living in the Community

The participants were asked to rank the life satisfaction characteristics of living closer to a landfill and not living close to a landfill from the scale of (1) satisfied, (2) somewhat satisfied or (3) not satisfied as shown in Table 5. Table 5 also shows the *t*-test analysis which was performed to understand the significant differences between both communities (Appendix D).

The results show that the participants living closer to the landfill site are less satisfied with the variables posed in this study than the participants living far from the landfill site. Figure 12 shows the graphical comparison of the participants’ views on how satisfied they are living in both communities.

The results show that the differences between the two communities were found to be statistically significant for five out of seven variables, that is for life in general; personal health condition; neighbourhood compared to others; community as a place to live in and perceived neighbourhood change. Palmiotto et al. [8] showed that residents living closer to the landfill experience higher forms of odour annoyance and the residents are concerned about the landfill impacts on the environment and their health. In addition, children living closer to landfills experience increased methane and *methanobrevibacter smithii* in their intestinal microbiota which caused serious health challenges and unrest in the community [60]. The study conducted in Nant-y-Gwyddeon landfill in South Wales showed that residents living in close proximity to landfill complain of odour and increased rate of congenital malformation [61].

## 4. Conclusions

The study on health and environmental impacts of landfill sites on humans has generated mixed reactions among scholars, therefore, constitutes a complex study. This study evaluated the health and environmental effects of Thohoyandou landfill on the residents living closer to the landfill, which integrates different factors like waste disposal, air and dust pollution, location of the landfill, water and noise pollution, fear of future health, property value, mosquitoes and rodent’s pollution, life in general in the community, etc.

This study concludes that the residents living closer to the landfill sites are at higher health and environmental risks when compared to those living far away from the landfill sites. However, the landfill associated problems have helped the community living closer to the landfill to be more conscious and educated on environmental pollution. The health risk associated with landfill pollutants in this study shows that proper landfill management is very essential. Landfills should be located far away from residential houses and institutions to avoid certain health and environmental related risks.

## Figures and Tables

**Figure 1 ijerph-16-02125-f001:**
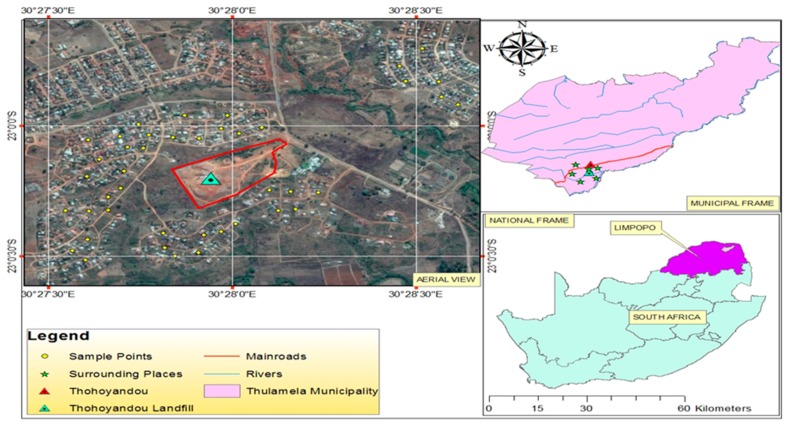
Thohoyandou landfill site and nearby residents.

**Figure 2 ijerph-16-02125-f002:**
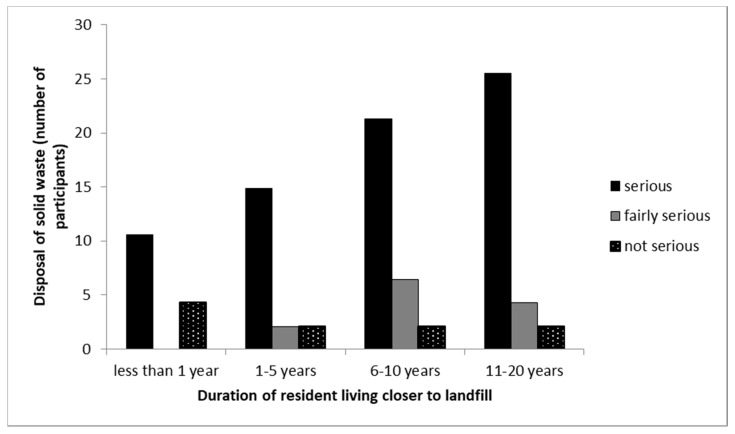
Cross-tabulation between the duration residents that lived in the CL community and the seriousness of solid waste disposal. Note that the no bar section in the figure indicates there was no response by the participant.

**Figure 3 ijerph-16-02125-f003:**
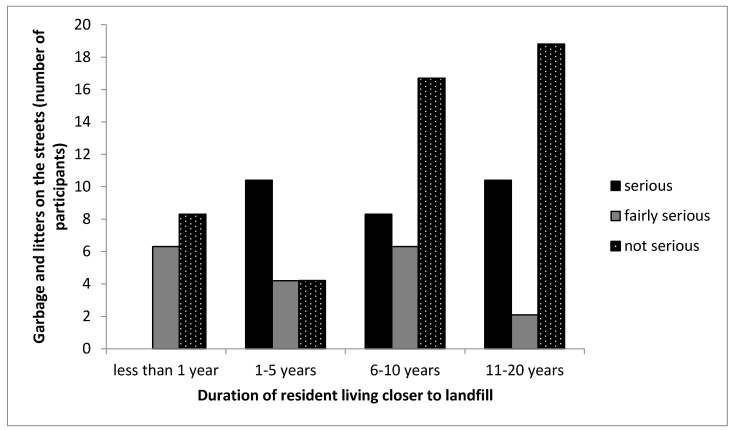
Cross-tabulation between the duration participants lived in the CL community and the seriousness of garbage and litter on the street. Note that the no bar section in the figure indicates there was no response by the participant.

**Figure 4 ijerph-16-02125-f004:**
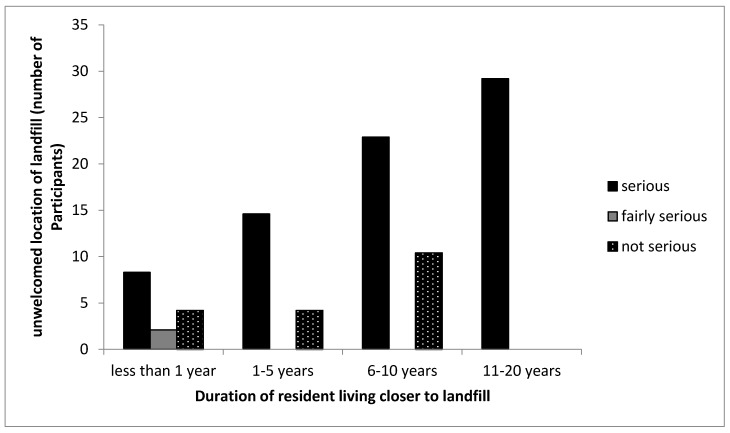
Cross-tabulation between the duration participants lived in the CL community and the seriousness of the unwelcomed location of the landfill. Note that the no bar section in the figure indicates there was no response by the participant.

**Figure 5 ijerph-16-02125-f005:**
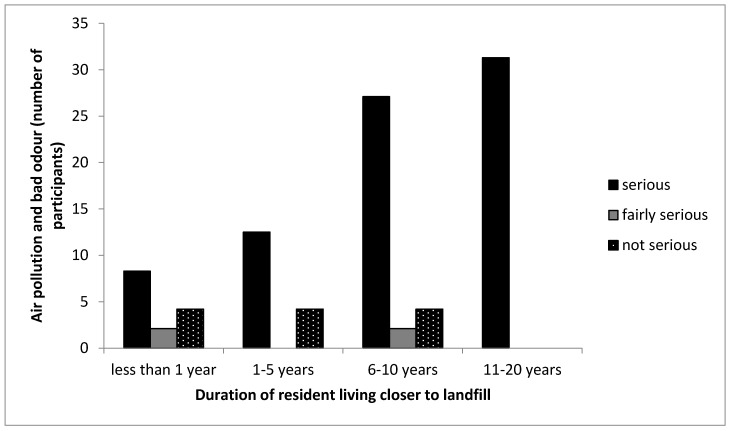
Cross-tabulation between the duration participants lived in the CL community and the seriousness of air pollution and bad odour. Note that the no bar section in the figure indicates there was no response by the participant.

**Figure 6 ijerph-16-02125-f006:**
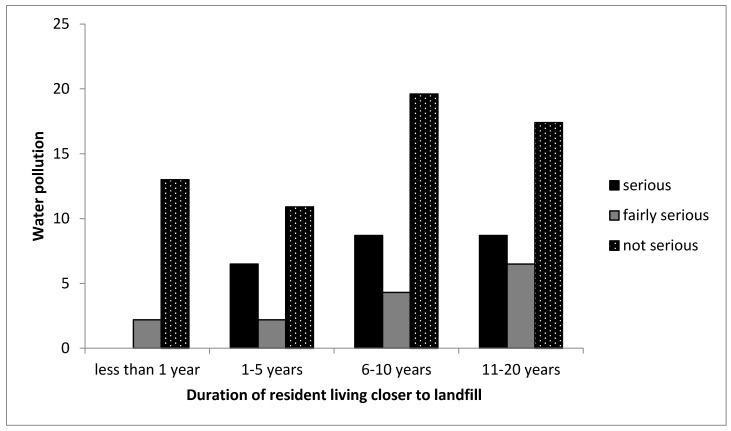
Cross-tabulation between the duration participants lived in the CL community and the seriousness of water pollution. Note that the no bar section in the figure indicates there was no response by the participant.

**Figure 7 ijerph-16-02125-f007:**
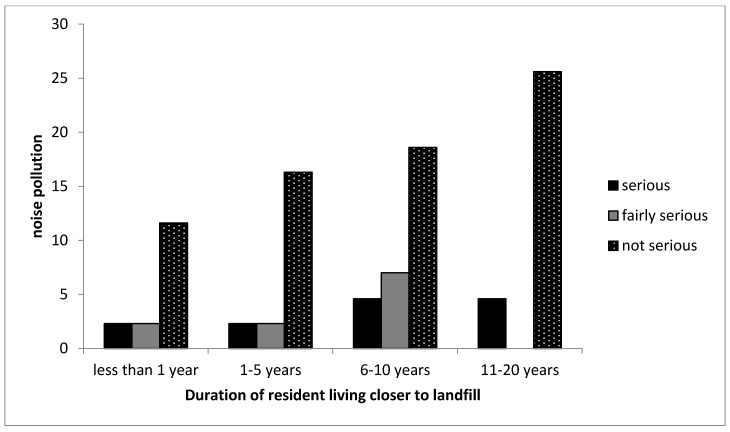
Cross-tabulation between the duration participants lived in the CL community and the seriousness of noise pollution. Note that the no bar section in the figure indicates there was no response by the participant.

**Figure 8 ijerph-16-02125-f008:**
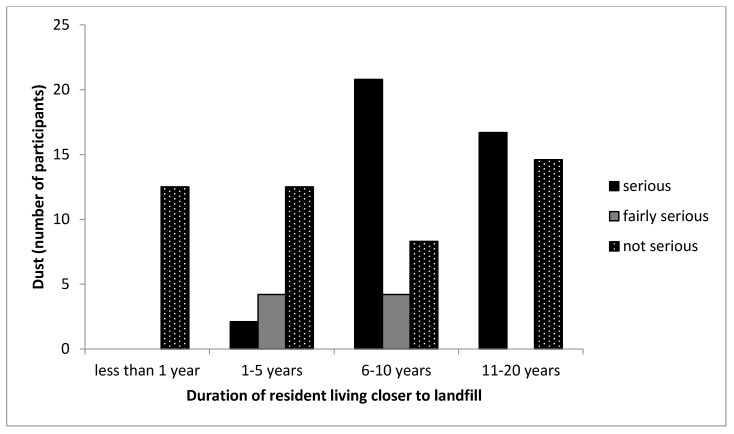
Cross-tabulation between the duration participants lived in the CL community and the seriousness of dust pollution. Note that the no bar section in the figure indicates there was no response by the participant.

**Figure 9 ijerph-16-02125-f009:**
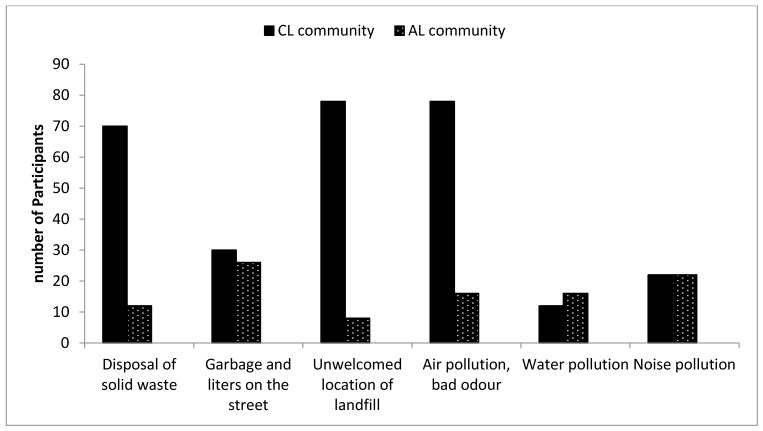
Comparison between both communities showing the seriousness of each landfill characteristics.

**Figure 10 ijerph-16-02125-f010:**
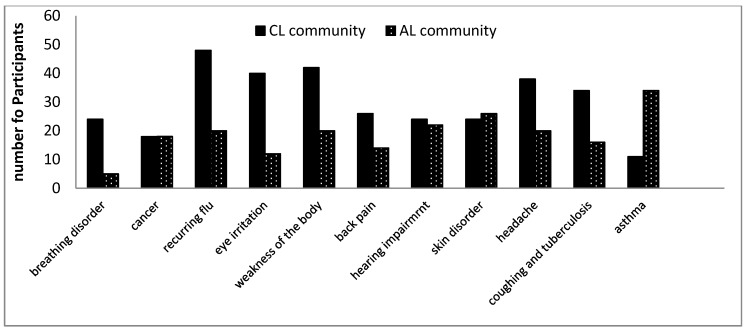
Comparison between both communities showing how frequent the illnesses impact them.

**Figure 11 ijerph-16-02125-f011:**
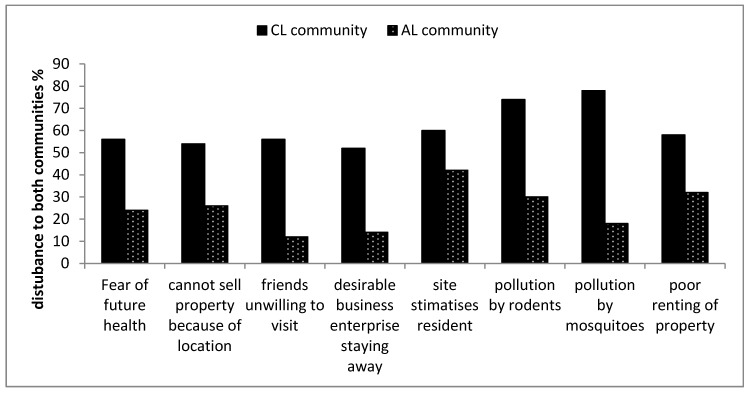
Comparison between both communities showing how disturbing these external factors were to them.

**Figure 12 ijerph-16-02125-f012:**
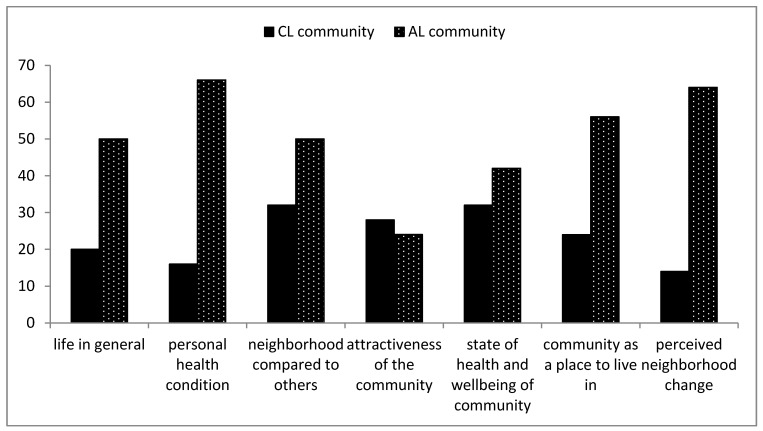
Comparison between both communities showing participants’ satisfaction.

**Table 1 ijerph-16-02125-t001:** Social and demographic characteristics of respondents.

	Living away from Landfill (AL)		Living Closer to Landfill (CL)	
	Number	Percentage %	Number	Percentage %
**Gender**				
Male	23	46	21	42
Female	27	54	29	58
Total	50	100	50	100.0
**Age**				
11–20 years	-	-	3	6
21–30 years	29	58	16	32
31–40 years	13	26	16	32
41–50 years	-	-	8	16
51 and above	8	16	6	12
Did not tell	-	-	1	2
Total	50	100.0	50	100
**Employment**				
Full-time employment	4	8	8	16
Part-time	11	22	11	22
Self-employed	13	26	7	14
Unemployed	22	44	20	40
Did not tell	-	-	4	8
Total	50	100.0	50	100.0
**Educational attainment**				
No formal education	4	8	5	10
Primary	12	24	6	12
High school	15	30	15	30
Tertiary education	19	38	20	40
Did not tell	-	-	4	8
Total	50	100.0	50	100.0
**Duration of time of living in the community**				
Less than 1 year	7	14	7	14
1–5 years	7	14	9	18
6–10 years	8	16	16	32
11–20 years	8	16	15	30
Above 20 years	20	40	2	4
Did not tell	-	-	1	2
Total	50	100.0	50	100.0

**Table 2 ijerph-16-02125-t002:** Respondents’ rating of the significance of environmental problems in the community.

Characteristics	Living Closer to Landfill (CL)				Living away from Landfill (AL)				Significance
	**Serious *n* (%)**	**Fairly Serious *n* (%)**	**Not Serious *n* (%)**	**Did Not Tell *n* (%)**	**Serious *n* (%)**	**Fairly Serious *n* (%)**	**Not Serious *n* (%)**	**Did Not Tell**	
Disposal of solid waste (landfill)	35 (70)	8 (16)	5 (10)	2 (4)	6 (12)	12 (24)	31 (62)	1 (2)	0.00
Garbage and litter in the street	15 (30)	9 (18)	25 (50)	1 (2)	13 (26)	28 (56)	6 (12)	3 (6)	0.027
Unwelcome location of the landfill	39 (78)	9 (18)	1 (2)	1 (2)	4 (8)	1 (2)	45 (90)	-	0.00
Air pollution, bad odour	39 (78)	3 (6)	7 (14)	1 (2)	8 (16)	12 (24)	28 (56)	2 (4)	0.00
Water pollution	6 (12)	6 (12)	32 (64)	6 (12)	8 (16)	20 (40)	21 (42)	1 (2)	0.034
Noise pollution	11 (22)	7 (14)	29 (58)	3 (6)	11 (22)	26 (52)	11 (22)	2 (4)	0.017
Dust	20 (40)	5 (10)	24 (48)	1 (2)	2 (4)	15 (30)	30 (60)	3 (6)	0.002

**Table 3 ijerph-16-02125-t003:** Respondents’ rating of how these illnesses are reported by the participants in both communities.

Characteristics	Living Closer to Landfill (CL)				Living Away from Landfill (AL)				Significance
	Frequent *n* (%)	Fairly Frequent *n* (%)	Not Frequent *n* (%)	Did Not Tell *n* (%)	Frequent *n* (%)	Fairly Frequent *n* (%)	Not Frequent *n* (%)	Did Not Tell *n* (%)	
Breathing disorder	12 (24)	17 (34)	20 (40)	1 (2)	5 (10)	12 (24)	33 (66)	-	0.009
Cancer	9 (18)	5 (10)	34 (68)	2 (4)	9 (18)	6 (12)	33 (66)	2 (4)	0.899
Reoccurring flu	24 (48)	11 (22)	13 (26)	2 (4)	10 (20)	17 (34)	22 (44)	1 (2)	0.005
Eye irritation	20 (40)	17 (34)	12 (24)	1 (2)	6 (12)	10 (20)	33 (66)	1 (2)	0.00
Weakness of the body	21 (42)	7 (14)	19 (38)	3 (6)	10 (20)	8 (16)	32 (64)	-	0.008
Back pain	13 (26)	5 (10)	29 (58)	7 (6)	7 (14)	2 (4)	41 (82)	-	0.040
Hearing impairment	12 (24)	2 (4)	30 (60)	6 (12)	11 (22)	2 (4)	37 (74)	-	0.537
Skin disorder	12 (24)	1 (2)	31 (62)	6 (12)	13 (26)	4 (8)	32 (68)	1 (2)	0.813
Headache	19 (38)	13 (26)	18 (36)	-	10 (20)	27 (54)	13 (26)	-	0.610
Coughing and Tuberculosis	17 (34)	14 (28)	17 (34)	2 (4)	8 (16)	14 (48)	28 (56)	-	0.016
Asthma	11 (22)	3 (6)	29 (58)	7 (14)	17 (34)	15 (30)	17 (34)	1 (2)	0.022

**Table 4 ijerph-16-02125-t004:** Respondents’ ratings of how disturbing the external characteristics are observed by the participants living in both communities.

Characteristics	Living Closer to Landfill (CL)				Living Away from Landfill (AL)				Significance
	Disturbing *n* (%)	Fairly Disturbing *n* (%)	Not Disturbing *n* (%)	Did Not Tell *n* (%)	Disturbing *n* (%)	Fairly Disturbing *n* (%)	Not Disturbing *n* (%)	Did Not Tell *n* (%)	
Fear of future health	28 (56)	10 (20)	3 (6)	9 (18)	12 (24)	9 (18)	28 (56)	1 (2)	0.00
Cannot sell the property because of location	27 (54)	11 (22)	4 (8)	8 (16)	13 (26)	2 (4)	33 (66)	2 (4)	0.00
Friends unwilling to visit	28 (56)	10 (20)	6 (12)	6 (12)	6 (12)	9 (18)	35 (70)	-	0.00
Desirable business enterprise staying away	26 (52)	8 (16)	6 (12)	10 (20)	7 (14)	6 (12)	33 (66)	4 (8)	0.00
Site stigmatizes resident	30 (60)	7 (14)	5 (25)	8 (16)	21 (42)	9 (18)	18 (36)	2 (4)	0.003
Pollution by rodents	37 (74)	7 (14)	2 (4)	4 (8)	15 (30)	-	35 (70)	-	0.00
Pollution by mosquitoes	39 (78)	6 (12)	1 (2)	4 (8)	9 (18)	12 (24)	27 (54)	2 (4)	0.00
Poor renting of property	29 (58)	8 (16)	5 (10)	8 (16)	16 (32)	2 (4)	29 (58)	3 (6)	0.00

**Table 5 ijerph-16-02125-t005:** Respondents’ ratings of how disturbing the external characteristics are as observed by the participants living in both communities.

Characteristics	Living Closer to Landfill (CL)				Living Away from Landfill (AL)				Significance
	Satisfied *n* (%)	Somewhat Satisfied *n* (%)	Unsatisfied. *n* (%)	Did Not Tell *n* (%)	Satisfied *n* (%)	Somewhat Satisfied *n* (%)	Unsatisfied *n* (%)	Did Not Tell *n* (%)	
Life in general	10 (20)	15 (30)	23 (46)	2 (4)	25 (50)	6 (12)	19 (66)	-	0.029
Personal health condition	8 (16)	20 (40)	20 (40)	2 (4)	33 (66)	1 (2)	15 (30)	1 (2)	0.000
Neighbourhood compared to others	16 (32)	15 (30)	15 (30)	4 (8)	25 (50)	20 (40)	5 (10)	-	0.015
Attractiveness of the community	14 (28)	10 (20)	21 (42)	5 (10)	12 (24)	20 (40)	18 (36)	-	0.834
State of health and wellbeing of participants	16 (32)	9 (18)	22 (44)	3 (6)	21 (42)	9 (18)	19 (38)	1 (2)	0.365
Community as a place to live in	13 (24)	17 (4)	18 (60)	2 (12)	28 (56)	12 (24)	10 (20)	-	0.005
Perceived neighbourhood change	7 (14)	11 (22)	30 (60)	2 (4)	32 (64)	13 (26)	5 (10)	-	0.00

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
