# Peer review of "Health and Environmental Risks of Residents Living Close to a Landfill: A Case Study of Thohoyandou Landfill, Limpopo Province, South Africa"

_ijerph, 2019, doi:10.3390/ijerph16122125_

Round 1

Reviewer 1 Report

Introduction

The theoretical background needs to be enhancing in your introduction or you can add a literature review. (Insufficient literature review.)

Materials and Methods. Concerning the methodology of the study. A lot of questions arise from the research design.  The organization and comprehensibility of the content in this section are very poor, please explain more in detail. The way the authors conducted the study is not very clear. The author must have in mind that the methodology section must have a very good redaction for the readers to have all the elements that lets them replicate the same procedure somewhere else.

Results.

Discussion. This section is poorly organized and the data have not discussed in a proper scientific manner.

This section is not supported by the analysis. Need to discuss with the literature review.    

The discussion part would need to have more references to the theory referred to in the theoretical background (Introduction section), to indicate how the results respond to the research questions and can be explained.

Conclusions. The Conclusion section should be more extensive and summarize the obtained results. In your conclusions, please discuss the implications of your research,

Others.

Figures. Title needs to be improved.  The titles must be short

Author Response

Thanks for the review and corrections, all corrections have been addressed in the manuscript

Reviewer 2 Report

In this paper the Authors assess the perception of residents living close and far from Thohoyandou landfill (South Africa) about environmental related issues, health problems and life satisfaction in these communities, through a questionnaire. 

This study is interesting and roughly well written, even if there some points that need to be reviewed:

1)    there are several typos that need to be corrected;

2)    the introduction must be implemented; in fact the Authors don’t deal with biological risk: see doi:  10.3390/ijerph13070631, doi:10.1183/09031936.03.00059702

3)    in paragraph 3.2 the Authors state that “24% of participants living closer to the landfill site indicated that breathing disorder was frequent, while 10% of the participants living far away fromthe landfill indicated the same problem. However, most of the participants living closer to the landfill(40%) indicated that breathing disorder had not been frequent. Similarly, 66% of the participantsliving far from the landfill indicated that breathing disorder had not been frequent.” Thus the Authors affirm: “therefore, theparticipants who have breathing disorder in the CL (participants that live closer to the landfill) and AL (participants that live far from the landfill) communities could have contracted itfrom different sources not necessarily from the landfill, since most of the participants had notexperienced the illness before”. This doesn’t make any sense, and so this statement must be removed.

4)    In the same paragraph the Authors must examine deeply the correlation between the presence of other risks rising from landfill (such as truck and bulldozer’s engine exhausts) and illnesses; see doi: 10.3390/ijerph121012977, doi: 10.1097/00130832-200410000-00003, doi: 10.1056/NEJMoa071535, PMID: 17291408 

Author Response

Thanks for the review and corrections. All corrections have been applied to in this manuscript

Round 2

Reviewer 1 Report

Authors have made all the changes

Author Response

Thanks, we have adhered to the your comments

Reviewer 2 Report

The revised manuscript appears improved, but there are some points that need to be reviewed:

1)      there are still typos that need to be corrected;

2)      in the introduction the Authors deal with biological risk very marginally; they must implement this part;

3)      Authors must examine deeply the correlation between the presence of other risks rising from landfill (such as truck and bulldozer’s engine exhausts) and illnesses (not only noise pollution, that is irrelevant in front of the biological risk and engine exhausts).

Author Response

Thanks for the comments, we have affected all changes

Round 3

Reviewer 2 Report

The Authors have significantly improved the manuscript, therefore I deem that this paper could be published in the present form.